# Gastric Cancer Risk in Association with Underweight, Overweight, and Obesity: A Systematic Review and Meta-Analysis

**DOI:** 10.3390/cancers15102778

**Published:** 2023-05-16

**Authors:** Narges Azizi, Moein Zangiabadian, Golnoosh Seifi, Afshan Davari, Elham Yekekhani, Seyed Amir Ahmad Safavi-Naini, Nathan A. Berger, Mohammad Javad Nasiri, Mohammad-Reza Sohrabi

**Affiliations:** 1School of Medicine, Tehran University of Medical Sciences, Tehran 1416634793, Iran; 2Endocrinology and Metabolism Research Center, Institute of Basic and Clinical Physiology Sciences, Kerman University of Medical Sciences, Kerman 7616913555, Iran; 3Basic and Molecular Epidemiology of Gastrointestinal Disorders Research Center, Research Institute for Gastroenterology and Liver Diseases, Shahid Beheshti University of Medical Sciences, Tehran 1985717443, Iran; 4Case Comprehensive Cancer Center, Case Western Reserve University, Cleveland, OH 44106, USA; 5Department of Microbiology, School of Medicine, Shahid Beheshti University of Medical Sciences, Tehran 1985717443, Iran; 6Community Medicine Department, School of Medicine, Shahid Beheshti University of Medical Sciences, Tehran 1983963113, Iran; 7Social Determinants of Health Research Center, Shahid Beheshti University of Medical Sciences, Tehran 1983963113, Iran

**Keywords:** stomach neoplasms, body mass index, overweight, obesity, risk factors, systematic review

## Abstract

**Simple Summary:**

Gastric cancer is the fifth most prevalent cancer in the world. The relationship between gastric cancer and underweight, overweight, and obesity is not fully understood yet. Obesity is a modifiable factor that has a positive association with some cancers. This systematic review and meta-analysis aimed to investigate the association between gastric cancer risk and abnormal body mass index, as an indicator of abnormal weight. Based on our results, obesity and overweight increase the risk of gastric cancer, while underweight is not associated. These findings can help policymakers and healthcare providers to make preventive strategies for controlling obesity and overweight.

**Abstract:**

This study aimed to investigate the risk of gastric cancer (GC) in abnormal body mass index (BMI) groups. A systematic search was carried out on Embase, PubMed/Medline, and Scopus from January 2000 to January 2023. The pooled risk ratio (RR) with a 95% confidence interval (CI) was assessed using a random-effect model. Thirteen studies with total of 14,020,031 participants were included in this systematic review. The pooled RR of GC was 1.124 (95% CI, 0.968–1.304, I^2^: 89.08%) in underweight class, 1.155 (95% CI, 1.051–1.270, I^2^: 95.18%) in overweight class, and in 1.218 (95% CI, 1.070–1.386, I^2^: 97.65%) obesity class. There is no difference between cardia and non-cardia gastric cancer, while non-Asian race and female gender have higher risk of cancer, as Meta-regression of obesity and overweight classes showed. These findings suggest that there is a positive association between excess body weight and the risk of GC, with a higher impact in women than men and in non-Asian than Asian populations. Since abnormal weight is tied to various diseases, including GC, healthcare experts, and policymakers should continue interventions aiming to achieve a normal BMI range.

## 1. Introduction

Gastric cancer (GC) is the fifth most prevalent cancer in the world and accounts for one million one hundred thousand new cases and seven hundred sixty-nine thousand deaths in 2020 [1]. Although GC incidence has declined over the past five years, it ranks the third cancer-related cause of death [2]. Cardia and non-cardia cancers are two entities of GC showing different trends [3]. The incidence of gastric cardia cancer has been increasing in recent decades, especially in developed countries [4]. However, non-cardia is the most commonly diagnosed type of gastric cancer worldwide, accounting for 82% of all cases [4,5]. Non-cardia incidence is higher in Eastern/Central Asia, and parts of Eastern Europe than in Western Europe and Northern America [5,6]. In the North American population, non-cardia incidence shows an age-specific trend and is decreasing in older adults [7]. This difference in incidence trends may be due to the underlying mechanism of carcinogenesis. Decreased incidence of gastric non-cardia cancer in several regions may be related to hygiene, H.Pylori eradication, lower intake of salt and alcohol, and higher intake of fresh fruit and vegetables [8,9,10,11]. On the other hand, the increased incidence of gastric cardia cancer may be related to obesity, smoking, and gastroesophageal reflux (GERD) although this relation has not been fully understood yet [9]. The incidence of both non-cardia and cardia cancers is higher in men than women [12]. This difference might be due to the higher prevalence of some risk factors such as smoking in men, as well as some protective factors such as reproductive hormones in women [12,13].

Over the past three decades, obesity rates have risen progressively, with a three-fold increase reported by the World Health Organization (WHO) since 1975 [14]. A recent study of the global burden of disease in one hundred ninety-five countries found that deaths and disability-adjusted life years (DALYs) attributable to high body mass index (BMI) had nearly doubled from 1990–2017 [15]. Obesity is associated with diabetes mellitus, cardiovascular diseases, hypertension, and different cancers including esophageal, endometrial, kidney, hepatocellular, pancreatic, colorectal and post-menopausal breast cancer [4,16,17]. The accumulation of fat tissue in different organs plays a key role in abnormal cell function, which may result in various chronic diseases such as cancer. Excess body fat, especially abdominal fat, is associated with insulin resistance. Since insulin and Insulin-like Growth Factor-I (IGF-I) have crucial roles in cell proliferation, alteration of insulin resistance can cause apoptosis impairment and tumorigenesis [18]. Additionally, pro-inflammatory cytokines such as leptin and adiponectin, which are produced in response to obesity, can trigger pro-inflammation and stimulate cancer development. Conversely, adiponectin is an apoptosis-inducing factor that is reduced in individuals with obesity in response to increased production of pro-inflammatory cytokines [19]. GERD, a probable risk factor for gastric cardia cancer, is also more prevalent in patients with obesity [20,21]. On the other hand, high consumption of red and processed meat and higher salt intake, which may increase the risk of obesity, are linked with a higher risk of GC [22,23,24,25]. Mentioned mechanisms can explain some extent of the obesity-cancer relation, and further research into obesity as a modifiable risk factor is essential for cancer prevention.

During recent years, several systematic reviews have assessed the relationship between GC and BMI though they have yielded variable results [26,27,28,29,30,31]. Among them, a meta-analysis in 2008 suggested obesity and GC have no significant association [26], while other analyses suggested a positive relationship between BMI and gastric cardia cancer [28,29,31]. Interestingly, a recent review by Bae et al. suggested overweight or obesity is a protective factor for GC in Asian men [30]. There was a moderate heterogeneity among articles, and some of the systematic reviews enrolled studies with duplicated populations. Nevertheless, controversies exist among studies, and this systematic review and meta-analysis aims to investigate the association of GC with abnormal BMI. Taking a consensus on the effects of obesity, overweight, and underweight on different diseases can influence public health strategies and governments policies to counter the obesity epidemic. 

## 2. Materials and Methods

This study was carried out based on Preferred Reporting Items for Systematic reviews and Meta-Analyses [32], the PRISMA statement [33], and was registered in the International Prospective Register of Systematic Reviews, PROSPERO under the registration number: CRD42021256634. 

### 2.1. Search Strategy

We conducted a systematic search in three databases including Embase, Pubmed/Medline, and Scopus for studies reporting the association between GC and BMI from 1 January 2000 to 1 January 2023. We selected original studies with cohort methodology written in English language from initial pool by following MeSH terms: ‘Stomach neoplasms’, ‘Body Weight’, ‘Body Mass Index’, ‘Obesity’, ‘Lifestyle’, ‘Demography’, ‘Social Factors’, ‘Socioeconomic Factors’, ‘Sociodemographic factors’, ‘Metabolic Syndrome’, and ‘Risk Factors’ (Appendix A). In addition, we manually searched the citations, related papers, and references of selected articles.

### 2.2. Study Selection

All papers retrieved were imported into EndNote X8 (Thomson Reuters, Toronto, ON, Canada), and studies meeting the following criteria were included: I. Cohort studies published between January 2000 to January 2023. II. Patients diagnosed with gastric adenocarcinoma (cardia or non-cardia). III. Patients aged over 20 years. IV. Papers providing data on GC incidence in relation to patients’ BMI based on either the WHO (underweight < 18.5, normal weight 18.5–24.9, overweight 25–29.9, and obese ≥ 30) or the Asian-Pacific category (underweight < 18.5, normal weight 18.5–22.9, overweight 23–24.9, and obese ≥ 25) in the main manuscript or Appendix A [34]. The excluded papers were unrelated, duplicated, non-English, abstract-only papers, theses, cross-sectional studies, case-control studies, case reports, case series, reviews, meta-analyses, clinical trials, editorials, and books. We also excluded papers about precancerous lesions, non-human subjects or in vitro, studies with an age lower than 20 years old, studies with a lack of exposed and unexposed population data related to outcome, studies with duplicated populations [35,36,37,38,39,40,41,42,43,44,45,46,47,48], and other types of studies except cohort studies. Regarding the selection method, two independent researchers selected studies based on title and abstract. In case of conflict, the third reviewer resolved the conflict through discussion with the initial reviewers. The same process was used for articles eligible for full-text review. 

### 2.3. Data Extraction

Two blinded reviewers independently extracted data from eligible studies and entered it into a designed data extraction form. In case of discrepancies between the two reviewers, a third reviewer extracted the data. This form included label, first author, publication year, study location, study design, baseline characteristics (age and gender), type and number of the outcome, baseline BMI, follow-up time, definition of exposed and unexposed population, and the number of exposed and unexposed population.

### 2.4. Quality Assessment

Two independent reviewers assessed the quality of the included studies using the Newcastle–Ottawa scale [49]. The scale contained three main parts: selection, comparability, and outcome. The first part was population selection, which contained the representative of the exposed cohort, selection of the non-exposed cohort, the ascertainment of exposure, and demonstration that outcome of interest was not present at start of study. The second part evaluated comparability based on the design or analysis. The third part contained the outcome assessment, long enough follow-up, and adequacy of follow-up of cohorts. The studies were categorized as poor (up to 3 scores), fair (4–6 scores), and good (7–9 scores) based on their quality, and only those rated fair and good were included in the meta-analysis.

### 2.5. Statistical Analysis

Statistical analyses were performed using Comprehensive Meta-Analysis software, version 3.7 (Biostat Inc., Englewood, NJ, USA). The pooled risk ratios (RRs) with 95% CI were assessed. The random-effects model was used because of the estimated heterogeneity of the true effect sizes. The between-study heterogeneity was assessed by Cochran’s Q test and the I^2^ statistic. I^2^ values of more than 50% were considered high heterogeneity [50]. Subgroup and subset analysis were performed to compare the role of race (between-studies), gender (within-studies), and type of GC (within-studies) in statistical heterogeneity and final effect size. These three pre-specified measures were known as possible sources of heterogeneity. Sensitivity analysis with one out remove method was done to determine if any particular study disproportionately impacts the overall results. Subgroup and subset analyses were performed separately for cardia and non-cardia cancers to compare the role of race (between-studies) and gender (within-studies) in the two subtypes of gastric cancer. Publication bias was also statistically evaluated using Begg’s and Egger’s tests as well as the funnel plot (*p* value < 0.05 was considered indicative of statistically significant publication bias and funnel plot asymmetry also suggested bias) [51]. 

## 3. Results

Figure 1 displays the flow diagram of study selection based on PRISMA [52]. We identified thirty thousand two hundred and forty-nine papers through databases (Embase, Pubmed/Medline, and Scopus) and citation searching, and screened thirteen thousand two hundred and eighty-five papers after removing duplicates. First, we ruled out thirteen thousand two hundred and eight papers by title and abstract since their subject, exposure or outcome were irrelevant to our study. We assessed seventy-eight studies by full-text review and included studies with international WHO or Asian-Pacific BMI cut-off points. We also took into account some papers with the BMI classifications fairly equal to these cut-off values [53,54]. Finally, thirteen cohort studies were assessed by NOS and included in the analysis. Table 1. shows the characteristics of included studies and their multivariate adjustment in brief). (Appendix A shows full detail of adjusted factors). One study was conducted in six states of the USA, one study in ten European countries (Denmark, France, Germany, Greece, Italy, the Netherlands, Norway, Spain, Sweden, and the United Kingdom), one in four Asian countries (based on thirteen cohorts from China (two), Japan (eight), Korea (two), and Singapore (one)), one in Japan, two in China, four in Republic of Korea, and one study were conducted in Austria, Norway, and the UK (Appendix A).

### 3.1. Quality of the Included Studies

Table 2 displays the quality assessment for the Cohort studies included in this analysis [49]. Among the thirteen studies that met our inclusion criteria, eleven were rated as good quality and two as fair quality, while none were classified as poor quality. We included all thirteen studies in the final meta-analysis.

### 3.2. Patient Characteristics

A total of fourteen million twenty thousand and thirty-one participants were included in the analysis drawn from thirteen studies [53,54,55,56,57,58,59,60,61,62,63,64,65]. The mean age of patients was 51.57 years, and 33.57% of patients were male. The mean follow-up time was 8.41 years. Table 3. shows the defined categories of BMI in the included studies. 

### 3.3. GC and Abnormal BMI

Twelve cohort studies investigated the risk of GC in participants with overweight, eleven in patients with underweight and eleven in patients with obesity. As shown in Figure 2, our meta-analysis showed that the pooled RR of GC in overweight and obese patients was 1.155 (95% CI, 1.051–1.270, I^2^: 95.18%) and 1.218 (95% CI, 1.070–1.386, I^2^: 97.65%). Therefore, overweight and obesity could mildly increase the risk of GC. However, this number was insignificant in the underweight class (pooled RR: 1.124; 95% CI, 0.968–1.304, I^2^: 91.57%). As between-study heterogeneity was high, we conducted a one-leave-out sensitivity analysis, which revealed that exclusion of any individual study did not significantly affect the overall findings, except for the study by Lim et al., in which exclusion resulted in a significant association between BMI and GC risk among underweight participants (Appendix A). There was no evidence of publication bias (*p* > 0.05, Appendix A).

### 3.4. Subgroup and Subset Analysis

Subgroup analysis was performed for race and subset analysis was used for gender and type of cancer. As shown in Table 4, there were no significant relations between type of cancer among patients with underweight, overweight or obesity (Appendix A). Non-Asian participants with overweight or obesity were at significant risk of GC but this risk was not significant in Asian participants. On the other hand, overweight and obesity in females could significantly increase the risk of GC, but this relation was insignificant in males with overweight and inverted in males with obesity (the difference between genders was significant in obesity and insignificant in overweight). Non-Asian race had the most role for statistical heterogeneity in obesity. This heterogeneity in patients with underweight was because of females and cardia cancer. Appendix A displays the pooled RRs of cardia cancer. Appendix A show the pooled RRs of cardia and non-cardia cancer.

## 4. Discussion

This meta-analysis of thirteen cohort studies endeavors to shed light on the controversial association of GC with abnormal BMI classes. Our findings suggest obesity and being overweight can increase the risk of GC by 21% and 15%, respectively. Low BMI, however, does not seem to affect GC risk. It is important to note that high heterogeneity was observed and should be considered when interpreting the results. To investigate the source of heterogeneity, we conducted between-studies (race) and within-studies (gender) analyses to assess the interacting effects of demographics. The within-studies analysis is the preferred method to compare influential factors [66], and it has a clear advantage over previous studies using stratified meta-analysis [28].

Several meta-analyses investigated the effects of obesity and GC, but the results have been controversial [27,28,29,30,31,32]. While some reviews found that being overweight increases the risk of GC [27,28], others have found no relation [26,29]. An analysis even found a protective effect of excess body weight on GC in Asian adults [30]. The inconsistencies could be due to differences in participant characteristics, BMI cut-off points, and meta-analysis method (Appendix A). 

The risk of GC associated with obesity and overweight is twice as high in non-Asian subgroup compared to all studies (Obesity: 57% and overweight: 43%). A similar but lower increase was evident in two meta-analyses of prospective cohorts [27,28]. Yang et al. first addressed the different GC-BMI associations in Asians in 2009. They showed obesity/overweight was associated with a higher risk of GC in non-Asians, while this relationship was not evident in Asians [27]. The results were confirmed by Lin et al. [28], and pooled analysis of seven cohorts revealed the paradoxical protective effect of high BMI in Asian males [30]. Although our result verified race differences, it denied the protective effect of obesity in any subgroup. This shift of direction happened because of recently published studies on the Asian population [54,64,65]. 

One possible explanation for the lower role of BMI in GC risk among Asians is that other risk factors, such as Helicobacter pylori infection [67,68], genetics [69], and high-sodium diet [70] and other environmental elements, may play a more prominent role in the development of GC in this population. Furthermore, the lower association between BMI and GC in the Asian population, in the current study may be due to the fact that Asians tend to have a higher proportion of body fat and a lower proportion of muscle mass compared to other populations at the same BMI level [71]. In 2004, the World Health Organization reviewed the evidence showing that Asians have a higher risk of developing weight-related diseases at lower BMIs [72]. However, due to a lack of agreement among researchers, it did not establish distinct BMI cutoffs for this population. In recent years, further studies have led some groups to adopt different BMI and abdominal obesity thresholds for Asians and even for different Asian groups [73,74]. Research has shown that Asian individuals tend to accumulate weight around the central region of their bodies, and at a faster rate than other ethnic groups [75]. Studies have shown that even with adjustment to age, BMI, and total fat mass, Chinese and South Asian individuals have significantly higher levels of visceral adipose tissue compared to White individuals [76]. Therefore, to assess the relationship between GC and body composition, especially in the Asian population, it may be necessary to consider other patterns of fat distribution using indicators such as waist circumference and waist-to-hip ratio may be necessary.

The impact of screening programs on the relationship between BMI and GC in Asian and non-Asian populations is another possible factor that may contribute to the observed difference. Several Asian countries have implemented screening programs due to the high prevalence of gastric cancer in these areas [77]. These programs aim to detect early lesions and improve overall survival rates by enabling timely intervention [78]. The implementation of such programs may affect the relationship between BMI and gastric cancer. In light of this, future studies may need to report the impact of screening programs on the relationship between BMI and gastric cancer. 

Interestingly, we found that females with excess weight have a higher risk of GC than males. In contrast, two systematic reviews in 2013 and 2014 found that males with obesity are more susceptible to GC than females by performing a “stratified meta-analysis” [28,29]. The 95% CI range of effect size for males overlaps with the CI range in females in both studies [28,29], which may reject significant differences between genders [66]. Besides, some indirect epidemiological findings point to a higher impact of excess weight in females. Analysis of worldwide cancer burden revealed that increased BMI is responsible for cardia GC in 11.2% of females and 8.8% of males [79].

The GC incidence itself shows a sex-difference, indicating the role of gender-related factors in carcinogenesis [80]. On the one hand, estrogen can protect women from GC by affecting estrogen receptors, stimulating trefoil expression, and interacting with leptin [60,81]. On the other hand, obesity is associated with higher estrogen levels and is also associated with increased GC risk [60]. One should also consider different lifestyle habits between genders, including smoking, alcohol consumption, physical activity, and diet [82]. The sex difference in obesity-related GC remained unexplained and requires future explorations.

The characteristics of GC differ between cardia and non-cardia GC. However, it is currently unclear whether this difference also applies to the relationship between obesity and GC as we could not confirm it. Two meta-analyses of case-control and cohort studies found that overweight or obesity was associated with an increased risk of cardia GC [31,83] or non-cardia GC [84]. 

The leading risk factor for cardia type is the high level of acid secretion in GERD, followed by gastritis caused by H. pylori infection [2,85]. Non-cardia GC is mainly associated with H. Pylori infection and diet habits, including consumption of preserved food and salted fish [2]. Eight out of thirteen included studies reported the risk of GC without considering the two subtypes’ different characteristics [59,61]. Future studies should address this issue, especially the different effect of general obesity and central obesity on GC subtypes.

This study, similar to some other reviews, discovered carrying extra weight can lead to cancer [26,28,29]. Several mechanisms are proposed for the carcinogenesis of excess body weight. Obesity is a systemic disease that causes systemic inflammation, insulin resistance, and hormone dysregulation, including high leptin, low adiponectin, and high IGF-I. In addition, excess weight occurs due to an unhealthy diet, high consumption of red and processed meat and high salt intake as well as low consumption of fruits and vegetables for instance, a sedentary lifestyle, and high-risk behaviors [22,25,86,87]. Consequently, obesity co-occurs with other comorbidities, such as GERD, which can increase the risk of gastric cardia cancer [88]. However, since the biopsychosocial context varies among different races, carcinogenesis may differ.

We found that being underweight is not associated with GC risk, which is also evident in other cohort studies [59,62]. However, the sensitivity analysis showed the direction of the effect was significantly influenced by Lim et al. study [65]. In particular, being underweight may increase the GC risk by excluding this influential study. In addition, the interaction of body weight and cancer has a complex pattern, seeing that studies proposed U- or J-shaped BMI-GC interaction [54,62,89]. Further studies are warranted to confirm the direction of the effect.

Other explanations for the effect of BMI on the carcinogenesis of GC have been proposed. The span that a person’s BMI exceeds the normal range in life has a strong relationship with the risk of GC [44,64,89]. In particular, patients who exceeded normal BMI in early adulthood had at least three times higher risk for gastric and esophageal cancers [44]. In addition, it is also possible that the cut-off points used in the BMI classification cannot stratify the risk of malignancy and are not optimum points for the classification of different populations [72]. Further epidemiologic and in-vitro studies are needed to investigate how adiposity can interact with the carcinogenesis of GCs.

## 5. Limitations

One should be aware of the limitations that need to be addressed before interpreting the results. I. The high level of heterogeneity among studies, which requires cautious interpretation of the results since future studies may change the direction of effect. The heterogeneity may be explained in part by variations in different gender and ethnic populations. II. Several potential confounding factors, including physical activity, dietary intake, H. pylori infection, and screening programs, could influence the results of the included observational studies. The presence of screening programs in Asian countries with high GC prevalence may alter the association between GC and other factors such as BMI. This could introduce bias since many patients benefit from early lesion removal and thus may not be included in the GC group. A meta-analysis of individual patient data may be warranted in the future since a meta-analysis itself cannot adjust for the effect of confounding factors. III. All non-English results and precancerous outcomes were excluded, which can cause bias. IV. Different BMI classifications caused the exclusion of some studies from the meta-analysis and subgroup analysis. V. All included cohorts are conducted in developed countries in Northern America, Europa, and East Asia. Therefore, results should be used in similar settings and contexts, while studies from developing countries are recommended. VI. The time of GCs’ onset after starting overweight or obesity could not be assessed, which may be of great importance.

## 6. Conclusions

According to this study, obesity and overweight can increase the risk of GC, whereas there is no significant relationship between underweight and GC. The impact of excess body weight on GC risk is higher in women and non-Asian populations. However, high heterogeneity is evident in the association between abnormal weight and GC risk, which may be explained in part by variations in different gender and race populations. Some unexplained questions should be addressed in future basic and epidemiological research. Since abnormal weight is tied to various diseases, including GC, healthcare experts and policymakers should continue interventions aimed at achieving a normal BMI range.

## Figures and Tables

**Figure 1 cancers-15-02778-f001:**
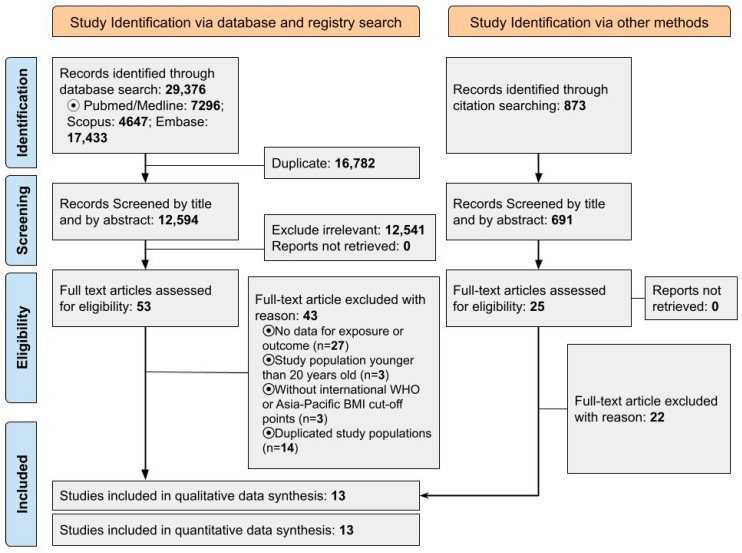
Flow chart of study selection for inclusion in the systematic review (PRISMA flow chart).

**Figure 2 cancers-15-02778-f002:**
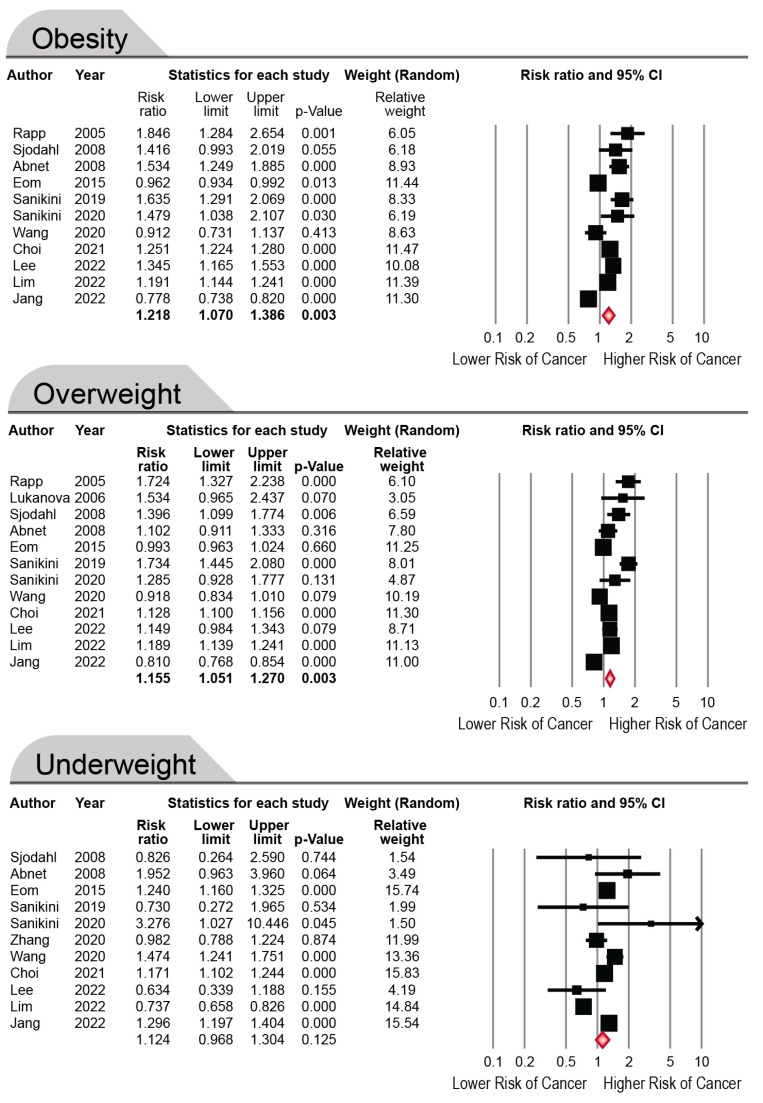
The Forest plot of included studies showing effect of different body mass index groups on gastric cancer risk. The random-effect model is used to adjust the study population size effects. Rapp 2005 [55]; Sjodahl 2008 [57]; Abnet 2008 [58]; Eom 2015 [59]; Sanikini 2019 [60]; Sanikini 2020 [61]; Wang 2020 [62]; Choi 2021 [63]; Lee 2022 [64]; Lim 2022 [65]; Jang 2022 [54]; Lukanova 2006 [56]; Zhang 2020 [53]. The black squares and lines represent the RRs and the CIs of the individual studies, respectively. The red diamond represents the pooled RR, and the outer edges of the diamond represent the CI.

**Table 1 cancers-15-02778-t001:** Characteristics and multivariate adjustment results of included cohort studies.

Study, Year, Location	Outcome	Ascertainment of Exposure	Adjustments	Result of Multivariate Adjusted Analysis
Rapp et al., 2005, Austria [55]	Gastric Adenocarcinoma	Measured by trained personnel	Age, year smoking, occupation	HR (95% CI): GC: Men: Overweight: 1.04 (0.73–1.47), obesity: 0.72 (0.40–1.33) Women: Overweight: 0.78 (0.51–1.20), obesity: 1.28 (0.76–2.15)
Lukanova et al., 2006, Japan [56]	Gastric Adenocarcinoma	Measured by trained personnel	Age, year, smoking	RR (95% CI): GC: Men: Overweight 1.36 (0.75–2.57) Women: Overweight: 0.53 (0.22–1.18)
Sjödahl et al., 2008, Norway [57]	Gastric Adenocarcinoma, Gastric Cardia Adenocarcinoma Gastric Noncardia Adenocarcinoma	Measured by trained personnel	Age, smoking, alcohol, meat, fish, fruit, veg	HR (95% CI): GC: Underweight: 0.7 (0.1–5.2); overweight: 1.0 (0.7–1.4), Obesity: 1.1 (0.7–1.8) GNCC: Underweight: 0.9 (0.1–6.7); overweight: 1.1 (0.7–1.6), obesity: 1.2 (0.7–2.1)
Abnet et al., 2008, Six USA states * [58]	Gastric Adenocarcinoma, Gastric Cardia Adenocarcinoma, Gastric Noncardia Adenocarcinoma	A self-reported questionnaire	Age, sex, smoking, alcohol, activity, edu, race	HR (95% CI): GCC: Underweight: 0.70 (0.10–5.06), overweight: 1.06 (0.79–1.41), Obese: 1.70 (1.22–2.36); GNCC: Underweight: 2.97 (1.38–6.39), overweight: 0.80 (0.61–1.04), Obese: 1.08 (0.78–1.50)
Eom et al., 2015, Korea [59]	Gastric Adenocarcinoma	Measured by trained personnel	Age, family history, meal, slat, alcohol, smoking	HR (95% CI): GC: Men: Underweight: 1.135 (1.051–1.226), overweight: 0.895 (0.864–0.927); Women GC: Underweight: 1.160 (1.010–1.333), overweight: 0.966 (0.906–1.030)
Sanikini et al., 2019, ten European countries ** [60]	Gastric Adenocarcinoma, Gastric Cardia Adenocarcinoma, Gastric Noncardia Adenocarcinoma	Measured by trained personnel	Age, center, smoking, edu., alcohol	HR (95% CI): GCC: Men: Overweight: 1.22 (0.86–1.75), obese: 0.94 (0.55–1.61); Women: Overweigh: 1.44 (0.85–2.43), obesity: 1.41 (0.70–2.83); GNCC: Men: Overweight: 1.13 (0.79–1.62); 1.03 (0.64–1.65); Women: Overweight: 0.96 (0.67–1.38), obesity: 1.31 (0.86–2.00)
Sanikini et al., 2020, UK [61]	Gastric Adenocarcinoma, Gastric Cardia Adenocarcinoma, Gastric Noncardia Adenocarcinoma	Measured by trained personnel	Age, sex, townsend index, smoking, edu	HR (95% CI): GCC: Overweight: 1.13 (0.71–1.82); obesity: 1.32 (0.79–2.21); GNCC: Overweight: 0.74 (0.45–1.23), obesity: 0.74 (0.42–1.32)
Zhang et al., 2020, China [53]	Gastric Adenocarcinoma	Measured by trained personnel	Age, gender, smoking, alcohol, family history, edu., fruit	HR (95% CI): GC: Underweight: 0.99(0.78–1.26), overweight or obesity: 1.06(0.73–1.55)
Wang et al., 2020 China [62]	Gastric Adenocarcinoma	Measured by trained personnel	Age, region, edu., marital status, income, alcohol, smoking, activity	HR (95% CI): GC: Underweight: 1.47 (1.22, 1.77), overweight: 0.94 (0.85, 1.06), obesity: 0.95 (0.76, 1.20)
Choi et al., 2021, Korea [63]	Gastric Adenocarcinoma	Measured by trained personnel	Smoking, alcohol, activity, income, age, parity, breastfeeding, contraceptive, HRT	HR (95% CI): GC in premenopausal women: Underweight: 1.12 (0.95–1.33), overweight: 0.96 (0.89–1.04), and obesity: 1.02 (0.94–1.11); GC in postmenopausal women: Underweight: 1.07 (1.00–1.14), overweight: 1.01 (0.99–1.04), and obesity: 1.03 (1.00–1.05)
Lee et al., 2022, Korea [64]	Gastric Adenocarcinoma	Measured by trained personnel	Age, sex, edu., smoking, alcohol, family history, activity, energy intake	HR (95% CI): GC (BMI at Baseline survey): Underweight: 0.67 (0.36–1.26), overweight 0.95 (0.81–1.11) and obesity: 1.08 (0.93–1.25)
Lim et al., 2022, Korea [65]	Gastric Adenocarcinoma	Measured by trained personnel	Age, sex, smoking, alcohol, exercise, income, DM, HTN, DLP	HR (95% CI): GC: Underweight: 1.15 (1.03–1.29), overweight 0.98 (0.93–1.02) and obesity: 1.03 (0.98–1.07)
Jang et al., 2022, thirteen cohorts from four Asian countries *** [54]	Gastric Adenocarcinoma, Gastric Cardia Adenocarcinoma, Gastric Noncardia Adenocarcinoma	Measured by trained personnel	Age, sex, country, smoking, alcohol	HR (95% CI): GC: Underweight: 1.15 (1.05–1.25), overweight: 1.01 (0.94–1.08), Obese: 1.12 (1.03–1.22); GCC: Underweight: 0.89 (0.58–1.38), overweight: 1.16 (0.86–1.57), Obese: 0.94 (0.62–1.43); GNCC: Underweight: 1.22 (1.10–1.35), overweight: 0.97 (0.89–1.05), Obese: 1.09 (0.98–1.21)

Footnote: * six U.S. states including California, Florida, Louisiana, New Jersey, North Carolina, and Pennsylvania) and two metropolitan areas (Atlanta, Georgia, and Detroit, Michigan. ** Ten European countries including Denmark, France, Germany, Greece, Italy, Norway, Spain, Sweden, the Netherlands, and the United Kingdom. *** Four Asian countries including China, Japan, Korea, and Singapore. GC; gastric cancer; GCC: gastric cardia cancer; GNC: gastric non-cardia cancer; DM: diabetes mellitus; HTN: hypertension; DLP: dyslipidemia; edu.: education; HR: hazard ratio; RR: risk ratio.

**Table 2 cancers-15-02778-t002:** Quality of included studies according to Newcastle-Ottawa scale ^a^.

Reference	1	2	3	4	5	6	7	8	Total Score
Rapp et al. [55]	-	*	*	*	*	*	*	*	*******
Lukanova et al. [56]	*	*	*	*	*	*	*	*	********
Sjödahl et al. [57]	*	*	*	-	*	*	*	-	******
Abnet et al. [58]	-	*	-	*	*	*	*	-	*****
Eom et al. [59]	-	*	*	*	*	*	*	*	*******
Sanikini et al. [60]	*	*	-	*	*	*	*	*	*******
Sanikini et al. [61]	*	*	*	*	*	*	*	*	********
Zhang et al. [53]	*	*	*	-	**	*	*	*	********
Wang et al. [62]	*	*	*	*	*	*	*	*	********
Choi et al. [63]	-	*	*	*	*	*	*	*	*******
Lee et al. [64]	*	*	*	*	**	*	*	-	********
Lim et al. [65]	*	*	*	*	*	*	*	*	********
Jang et al. [54]	*	*	*	*	*	*	*	*	********

Footnote: ^a^ Each item can be scored a maximum of one star, except for item 5, which can be scored up to two stars. The maximum total score is 9 stars. 1: Representativeness of the exposed cohort; 2: Selection of the non-exposed cohort; 3: Ascertainment of exposure; 4: Demonstration that outcome of interest was not present at start of study; 5: Comparability of cohorts on the basis of the design or analysis; 6: Assessment of outcome; 7: Was follow-up long enough for outcomes to occur; 8: Adequacy of follow up of cohorts.

**Table 3 cancers-15-02778-t003:** Cohorts’ characteristics and body mass index (BMI) categories used for obesity classification.

References	Race	Cohort Members	Gender (Male %)	Mean Age (Years)	Follow-Up Time (Years)	BMI Categories (kg/m^2^)
Rapp et al. [55]	Non-Asian	145,931	54	42.2	9.93	Participants with normal weight (18.5–24.9), overweight (25–29.9), and obesity (≥30)
Lukanova et al. [56]	Non-Asian	68,786	48.6	46.1	8.2	Participants with normal weight (18.5–24.9), overweight (25–29.9), and obesity (≥30)
Sjödahl et al. [57]	Non-Asian	72,487	49	49.0	15.4	Participants with underweight (<18.5), normal weight (18.5–24.9), overweight (25–29.9), and obesity (≥30)
Abnet et al. [58]	Non-Asian	480,475	60	62.0	8	Participants with underweight (<18.5), normal weight (18.5–24.9), overweight (25–29.9), obesity (30–35), and extremely obesity (≥35)
Eom et al. [59]	Asian	2,176,501	63	46.4	11.3	Participants with underweight (<18.5), normal weight (18.5–22.9), overweight (23–24.9), and obesity (≥25) Asian-Pacific BMI cut-offs were used to define BMI subgroups
Sanikini et al. [60]	Non-Asian	391,474	36	51.3	14	Participants with underweight (<18.5), normal weight (18.5–24.9), overweight (25–29.9), and obesity (≥30)
Sanikini et al. [61]	Non-Asian	455,166	46.6	57.4	6.5	Participants with underweight (<18.5), normal weight (18.5–24.9), overweight (25–29.9), and obesity (≥30)
Zhang et al. [53]	Asian	3298	44	55.0	30	Participants with underweight (<18.5), normal weight (18.5–23.9), and overweight or obesity (≥24) Asian-Pacific BMI cut-offs were used to define BMI subgroups
Wang et al. [62]	Asian	508,362	41	51.5	8.95	Participants with underweight (<18.5), lower BMI (18.5–21.9), normal weight (22.0–24.9), overweight (25–29.9), and obesity (≥30)
Choi et al. [63]	Asian	6,272,367	0	56.2	7.2	Participants with underweight (<18.5), normal weight (18.5–22.9), overweight (23–24.9), obesity (25–29.9), and severely obesity (≥30) Asian-Pacific BMI cut-offs were used to define BMI subgroups
Lee et al. [64]	Asian	134,130	34	52.4	8.6	Participants with underweight (<18.5), normal weight (18.5–22.9), overweight (23–24.9), and obesity (≥25) Asian-Pacific BMI cut-offs were used to define BMI subgroups
Lim et al. [65]	Asian	2,757,017	74	42.5	6.78	Participants with underweight (< 18.5), normal weight (18.5–22.9), overweight (23–24.9), and obesity (≥ 25) Asian-Pacific BMI cut-offs were used to define BMI subgroups
Jang et al. [54]	Asian	554,037	45.3	54.4	14.9	Participants with underweight (<18.5), normal weight (18.5–23), overweight (23–27.5), and obesity (≥27.5) Asian-Pacific BMI cut-offs were used to define BMI subgroups.

**Table 4 cancers-15-02778-t004:** Subgroup analysis of race, gender, and type of cancer effect on gastric cancer in different body mass index categories.

Potential Factors	RR (CI 95%)	No of Studies	Heterogeneity *χ^2^*	*p* Value	I^2^%	Interaction *p* Value
**Obesity**						
Race	Subgroup analysis					
Asian Non-Asian	1.056 (0.900–1.238) 1.573 (1.389–1.780)	6 5	394.63 1.36	0.000 0.852	98.73 0.00	0.000
Gender	Subset analysis					
Male Female	0.983 (0.835–1.158) 1.554 (1.223–1.976)	4	7.09 12.93	0.069 0.005	57.70 76.80	0.002
Type of Cancer	Subset analysis					
Cardia Non-Cardia	1.318 (0.803–2.162) 1.182 (0.728–1.919)	5	32.52 69.61	0.000 0.000	87.70 94.25	0.759
All studies	1.218 (1.070–1.386)	11	425.83	0.000	97.65	-
Overweight						
Race	Subgroup analysis					
Asian Non-Asian	1.019 (0.914–1.137) 1.438 (1.209–1.711)	6 6	178.04 13.98	0.000 0.016	97.19 64.24	0.001
Gender	Subset analysis					
Male Female	1.052 (0.821–1.348) 1.462 (1.173–1.823)	5	24.15 15.09	0.000 0.005	83.44 73.50	0.052
Type of Cancer	Subset analysis					
Cardia Non-Cardia	1.292 (0.899–1.858) 1.107 (0.765–1.602)	5	23.77 59.18	0.000 0.000	83.17 93.24	0.559
All studies	1.155 (1.051–1.270)	12	228.43	0.000	95.18	-
**Underweight**						
Race	Subgroup analysis					
Asian Non-Asian	1.097 (0.940–1.281) 1.424 (0.742–2.732)	7 4	85.27 5.35	0.000 0.148	92.96 43.90	0.446
Gender	Subset analysis					
Male Female	1.829 (0.774–4.324) 1.137 (0.993–1.301)	4	13.46 2.58	0.004 0.460	77.71 0.00	0.284
Type of Cancer	Subset analysis					
CardiaNon-Cardia	0.958 (0.662–1.387) 1.611 (0.975–2.660)	5	1.69 7.78	0.792 0.100	0.00 48.60	0.102
All studies	1.124 (0.968–1.304)	11	91.57	0.000	89.08	-

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
