# Peer review of "Gastric Cancer Risk in Association with Underweight, Overweight, and Obesity: A Systematic Review and Meta-Analysis"

_cancers, 2023, doi:10.3390/cancers15102778_

Round 1

Reviewer 1 Report

The authors conducted an interesting systematic review and meta-analysis study based on 14 cohort studies, which included 13,585,589 participants. The study addressed the associations between abnormal body mass index and the risk of gastric cancer, both overall and by anatomical subtypes. However, some major concerns need to be responded, as follows:

Major comments:

[lines 50-51] “The incidence of gastric non-cardia decreased while cardia cancer increased in recent decades [3-5].” This statement is not completely accurate. While the incidence of cardia cancer has increased, particularly in developed countries, the incidence of non-cardia cancer is higher in East/Central Asia and Eastern Europe than in North America and Western Europe. Additionally, the incidence of non-cardia cancer was found to be age-dependent in the North American population.

It should be noted that the level of heterogeneity between studies is high in overweight and obese groups. This should also be described in words. [Lines 193-194]

3. Gender and ethnicity do not necessarily account for the high heterogeneity in this study.

One-leave-out analysis might help determine if any particular study disproportionately impacts the overall results.

The findings on gastric cancer (GC) risk in Asia require careful consideration and examination in the discussion section [Lines 250-257]. Firstly, the commonly used Asian-Pacific BMI categories may not be the best approach for categorizing BMI in the Asian population. (Please see the WHO Expert Consultation on the Appropriate Body-mass Index for Asian Populations and Its Implications for Policy and Intervention Strategies). Secondly, a recently published paper (September 2022) by Jang et al., using data from more than 500,000 East and Southeast Asians from the Asian Consortium Cohort study, found a positive U-shaped association between BMI and GC, which contradicts the results of this study.

Only five studies have data on anatomical cancer subsites. Therefore, conclusions about GC anatomical subsites are primarily based on Western studies (4). It would be advisable to evaluate the inclusion of the Jang et., al paper to lessen the non-cardia limitation and for the overall GC analysis.

Jang J, Lee S, Ko KP, et al. Association between Body Mass Index and Risk of Gastric Cancer by Anatomic and Histologic Subtypes in Over 500,000 East and Southeast Asian Cohort Participants. Cancer Epidemiol Biomarkers Prev.             2022;31(9):1727-1734. doi:10.1158/1055-9965.EPI-22-0051

While the limitations of physical activity and dietary intake as confounding variables were mentioned, the influence of Helicobacter pylori was not. H. pylori-positive patients have been found to have a higher likelihood of being obese, and obese individuals have a higher risk of H. pylori infection. Given that few observational studies account for this variable, it is important to mention and explain H. pylori as a limitation.

Minor comments

Please maintain consistency in the number of decimal places throughout the manuscript, figures, and tables. It is often advisable to use 3 decimal places for p-values and 2 decimal places for percentages. Please refer to the journal guidelines for further instructions.

Author Response

Dear Reviewer,

The authors want to appreciate the time and effort that the reviewers and editor provided to give concise feedback and comments for our manuscript entitled “Gastric Cancer Risk in Association with Underweight, Overweight, and Obesity: A Systematic Review and Meta-Analysis”. We improved the manuscript and corrected flaws using the editor and reviewers’ comments. Here is a point-by-point response to the reviewers’ comments and concerns (the address of change is presented at the end of the response, in the parenthesis). All changes are marked using the track change view of MS Word.

In brief, we added more details on the trends of subtypes of gastric cancer per reviewer comment in the Introduction (Introduction paragraph 1). In the Method section, we added the mentioned paper entitled "Association between Body Mass Index and Risk of Gastric Cancer by Anatomic and Histologic Subtypes in Over 500,000 East and Southeast Asian Cohort Participants" by Jang et., al. We also excluded two duplicated cohort studies by Kuriyama et al. 2005 and Hirabayashi et al. 2019 as they had already been included in Jang et al.'s study. Additionally, we carried out One-leave-out analysis to find out the source of heterogeneity, and added two supplementary tables based on gastric cardia and non-cardia cancers (Tables S3, and S4). We also removed tables S2, which provided data on pooled relative risk in Asian and non-Asian ethnicities stratified by gender, to avoid multiple comparisons. In addition, the Discussion section was reorganized and duplicate statements was removed. The whole manuscript was checked for grammar issues and was rewritten in some cases for better English comprehension.

Comment 1: “The incidence of gastric non-cardia decreased while cardia cancer increased in recent decades [3-5].” This statement is not completely accurate. While the incidence of cardia cancer has increased, particularly in developed countries, the incidence of non-cardia cancer is higher in East/Central Asia and Eastern Europe than in North America and Western Europe. Additionally, the incidence of non-cardia cancer was found to be age-dependent in the North American population.

R1-C1-Response: Thank you for reviewing this study and for bringing up this point. We updated the statement for increased accuracy as follows: “The incidence of gastric cardia cancer has been increasing in recent decades, especially in developed countries (4). However, non-cardia is the most commonly diagnosed type of gastric cancer worldwide, accounting for 82% of all cases (4, 5). Non-cardia incidence is higher in Eastern/Central Asia, and parts of Eastern Europe compared to Western Europe and Northern America (5, 6). In the North American population, non-cardia incidence shows an age-specific trend and is decreasing in older adults.” We also added relevant references to support these statements. (Address of changes: Introduction: Paragraph 1, Lines 59-66).

Comment 2: It should be noted that the level of heterogeneity between studies is high in overweight and obese groups. This should also be described in words.

R1-C2-Response: Thanks for pointing this out. To address the high level of heterogeneity between studies, we added the following statement: “As the between study heterogeneity was high for these analyses, one-leave-out analysis was done.” (Address of changes: Results: Part 3.3, Lines 246-248).

In addition, we added a statement in the conclusion and discussion to further emphasize this issue (4. Discussion, Line 287-290.)

Comment 3: Gender and ethnicity do not necessarily account for the high heterogeneity in this study.

R1-C3-Response: we appreciate the reviewer's feedback. The heterogenicity of effect size decreased in non-Asian and gender. The authors agree with the reviewer that this heterogenicity is not completely explained by variations in gender and ethnicity. However, it can play a role in this regard. As per reviewer comment we revised the discussion to declare this issue (5. Discussion, Line 379: “Variations in different gender and ethnicities population may explain some extent of this heterogenicity.”)

Comment 4: One-leave-out analysis might help determine if any particular study disproportionately impacts the overall results.

R1-C4-Response: The authors thank the reviewer for this comment. As the between study heterogeneity was high for these analyses, one-leave-out analysis was done and showed that any of studies did not change the final results except Lim et al. study that removing it from underweight analysis would result in significant association (Address of changes: Results: Part 3.3, Lines 247-251; Supplementary Figure S2; Discussion, Line 347).

Comment 5: The findings on gastric cancer (GC) risk in Asia require careful consideration and examination in the discussion section [Lines 250-257]. Firstly, the commonly used Asian-Pacific BMI categories may not be the best approach for categorizing BMI in the Asian population. (Please see the WHO Expert Consultation on the Appropriate Body-mass Index for Asian Populations and Its Implications for Policy and Intervention Strategies). Secondly, a recently published paper (September 2022) by Jang et al., using data from more than 500,000 East and Southeast Asians from the Asian Consortium Cohort study, found a positive U-shaped association between BMI and GC, which contradicts the results of this study.

R1-C5-Response: We would like to express our gratitude for this insightful comment.

According to the WHO Expert Consultation on Appropriate Body-Mass Index for Asian Populations and Its Implications for Policy and Intervention Strategies, the suggested categories for BMI are as follows: less than 18·5 kg/m2 underweight; 18·5–23 kg/m2 increasing but acceptable risk; 23–27·5 kg/m2 increased risk; and 27·5 kg/m2 or higher high risk." However, we could only extract primary data on this cut-off point from the Jang et al. study. Therefore, to include more primary data, we had to choose a cut-off point that was most similar to the cut-off points used in the included Asian studies. We acknowledged this as a limitation of our study in the discussion section. (Address: Discussion lines 380-381)

We included the Jang et al. paper in our study, and we appreciate your point because we could not find its full text during our last search. Jang's paper included 13 Asian cohorts, three of which were already duplicated in our included studies. To avoid including duplicated data, we excluded two Japanese studies: Kuriyama et al. 2005 (3pref. Miyagi cohort) and Hirabayashi et al. 2019 (JPHC1,2 cohorts). We reconducted the analyses and found that the results did not change meaningfully except for gastric cardia cancer in individuals with obesity and females with underweight. We also made changes to the PRISMA flow diagram, tables, figures, and other relevant sections of the study to include the Jang et al. 2022 paper and exclude the Kuriyama et al. 2005 and Hirabayashi et al. 2019 papers. Asian population considerations are described in the discussion section. (Address: Discussion lines 300-310)

Comment 6: While the limitations of physical activity and dietary intake as confounding variables were mentioned, the influence of Helicobacter pylori was not. H. pylori-positive patients have been found to have a higher likelihood of being obese, and obese individuals have a higher risk of H. pylori infection. Given that few observational studies account for this variable, it is important to mention and explain H. pylori as a limitation.

R1-C6-Response: The authors want to express their gratitude for reviewer’s precision. We added H.pylori along with other confounder in the limitations of our study (Discussion, Line 378).

Comment 7: Please maintain consistency in the number of decimal places throughout the manuscript, figures, and tables. It is often advisable to use 3 decimal places for p-values and 2 decimal places for percentages. Please refer to the journal guidelines for further instructions.

R1-C7-Response: Thanks for pointing this out. We used 3 decimal places for p-values and 2 decimal places for percentages. We also used 3 decimal places for the effect sizes and confidence intervals.

The authors want to again appreciate the time and effort that reviewers and editors put into evaluating this study. The comments improved the manuscript and analysis, and any further comments would be welcomed.

Bests,

Mohammad Javad Nasiri (mj.nasiri@hotmail.com)

Reviewer 2 Report

The topic of the manuscript is interesting.

However, the authors should:

1. Improve English 

2. Divide the results in cardiaGC and non cardiaGC, as they are different types of tumors, even if the results for non cardia GC are few 

3. The discussion should be reformulated and revised, making clear the points to discuss and rationally organising the concepts. Indeed, as it is presented now, it appears confused and it seems to the reader that the points are continuously repeated

Author Response

Dear Reviewer,

The authors want to appreciate the time and effort that the reviewers and editor provided to give concise feedback and comments for our manuscript entitled “Gastric Cancer Risk in Association with Underweight, Overweight, and Obesity: A Systematic Review and Meta-Analysis”. We improved the manuscript and corrected flaws using the editor and reviewers’ comments. Here is a point-by-point response to the reviewers’ comments and concerns (the address of change is presented at the end of the response, in the parenthesis). All changes are marked using the track change view of MS Word.

In brief, we added more details on the trends of subtypes of gastric cancer per reviewer comment in the Introduction (Introduction paragraph 1). In the Method section, we added the mentioned paper entitled "Association between Body Mass Index and Risk of Gastric Cancer by Anatomic and Histologic Subtypes in Over 500,000 East and Southeast Asian Cohort Participants" by Jang et., al. We also excluded two duplicated cohort studies by Kuriyama et al. 2005 and Hirabayashi et al. 2019 as they had already been included in Jang et al.'s study. Additionally, we carried out One-leave-out analysis to find out the source of heterogeneity, and added two supplementary tables based on gastric cardia and non-cardia cancers (Tables S3, and S4). We also removed tables S2, which provided data on pooled relative risk in Asian and non-Asian ethnicities stratified by gender, to avoid multiple comparisons. In addition, the Discussion section was reorganized and duplicate statements was removed. The whole manuscript was checked for grammar issues and was rewritten in some cases for better English comprehension.

Comment 1: Improve English

R2-C1-Response: We thank you for your review and comment on our study. We used an external scientific editor and reviewed the whole manuscript for flaws. All changes throughout the manuscript are highlighted using the track change view.

Comment 2: Divide the results in cardia GC and non-cardia GC, as they are different types of tumors, even if the results for non-cardia GC are few

R2-C2-Response: Thanks for pointing this out. In addition to providing data on gastric cancer subtypes in the main manuscript by subgroup analysis (Table 4), we added two supplementary tables provided data specifically on gastric cardia and non-cardia cancers. Due to the limited availability of data (only one Asian and four non-Asian studies that provided data on cardia and non-cardia cancers) we included these tables in the supplementary material. (Address: Supplementary file, Tables S3, and S4)

Comment 3: The discussion should be reformulated and revised, making clear the points to discuss and rationally organizing the concepts. Indeed, as it is presented now, it appears confused and it seems to the reader that the points are continuously repeated

R2-C3-Response: Thanks for your time and effort on reading and evaluating the manuscript. We reorganized the whole discussion, and removed many duplicated statements. Also, we asked an external reviewer to read the manuscript, and we corrected flaws in the whole manuscript.

The authors want to again appreciate the time and effort that reviewers and editors put into evaluating this study. The comments improved the manuscript and analysis, and any further comments would be welcomed.

Bests,

Mohammad Javad Nasiri (mj.nasiri@hotmail.com)

Round 2

Reviewer 1 Report

The revised manuscript was found to be much improved.

Minor comments:

248-252 This paragraph may be rewritten by the authors for clarity.

Although ethnicity and race are frequently used interchangeably, they are slightly different. Please ensure that the terminology used throughout the paper is consistent.

Because of the substantial heterogeneity among the studies and the possibility of studies influencing steering switches, as the authors stated, authors might consider lowering the tone of the conclusions. 

Check for spelling mistakes in the graphic abstract: inclusion

Author Response

Dear reviewer,

The authors want to appreciate the time and effort that the reviewers and editor provided to give concise feedback and comments for our manuscript entitled “Gastric Cancer Risk in Association with Underweight, Overweight, and Obesity: A Systematic Review and Meta-Analysis”. We improved the manuscript and corrected flaws using the editor’s and reviewers’ comments. Here is a point-by-point response to the reviewers’ comments and concerns (the address of change is presented at the end of the response, in the parenthesis). All changes are marked using the track change view of MS Word.

In brief, we used 'race' instead of 'ethnicity' as appropriate, lowered the tone of conclusion due to the heterogeneity among the studies, revised the graphical abstract spelling mistake, provided possible explanations for the lower role of BMI in GC risk among Asians, and improved Introduction and Discussion by adding mentioned points about cardia and not cardia GC and dietary factors.

Comment 1: 248-252 This paragraph may be rewritten by the authors for clarity.

R1-C1-Response: Thank you for bringing up this point. We rewrote it for clarity. We stated “As between-study heterogeneity was high, we conducted a one-leave-out sensitivity analysis, which revealed that exclusion of any individual study did not significantly affect the overall findings, except for the study by Lim et al., which exclusion resulted in a significant association between BMI and GC risk among underweight participants.”

Comment 2: Although ethnicity and race are frequently used interchangeably, they are slightly different. Please ensure that the terminology used throughout the paper is consistent.

R1-C2-Response: Thank you for bringing this to our attention. We revised the manuscript to ensure consistent terminology throughout, using 'race' instead of 'ethnicity' as appropriate.

Comment 3: Because of the substantial heterogeneity among the studies and the possibility of studies influencing steering switches, as the authors stated, authors might consider lowering the tone of the conclusions.

R1-C3-Response: Thank you for your insightful feedback. We added this statement to conclusion section: “High heterogeneity is evident in the association between abnormal weight and GC risk. Variations in different gender and race populations may explain some extent of this heterogeneity.” (Lines 391-393)

Comment 4: Check for spelling mistakes in the graphic abstract: inclusion.

R1-C4-Response: Thank you for your feedback. We revised the graphical abstract accordingly.

The authors want to again appreciate the time and effort that reviewers and editors put into evaluating this study. The comments improved the manuscript and analysis, and any further comments would be welcomed.

Bests,

Mohammad Javad Nasiri (mj.nasiri@hotmail.com)

Reviewer 2 Report

The english language and the whole manuscript has been improved.

Still I see some points to modify:

- regarding the association between BMI and race, Asian race seems to be less influenced by obesity and overweight in the manuscript. The asian populations are known to have the highest rate of GC in the world and there are also screening programs for GC. Please, comment on these two points: how the highest prevalence of GC overall can determine the lower role of BMI and how the available screening (Endoscopy, sierological tests...) could influence the prevalence of GC and therefore create a big bias in the evaluation of the role of BMI between asian and no asian populations. There are a lot of studies on these topics. It would be interesting to adjust the analysis, if possible.      

- There are several studies on the role of meat, salt and diet in general and GC. Please, add them in the introduction and in the discussion

- In the discussion: from the line 358 to 370 the authors talk about obesity in general. This part should belong to the introduction 

- Still I think that the discussion about cardia and not cardia GC is little and should be improved. Indeed the discussion should be implemented with studies on the association between obesity and MRGE and not-cardia GC, if possible

Author Response

Dear reviewer,

The authors want to appreciate the time and effort that the reviewers and editor provided to give concise feedback and comments for our manuscript entitled “Gastric Cancer Risk in Association with Underweight, Overweight, and Obesity: A Systematic Review and Meta-Analysis”. We improved the manuscript and corrected flaws using the editor’s and reviewers’ comments. Here is a point-by-point response to the reviewers’ comments and concerns (the address of change is presented at the end of the response, in the parenthesis). All changes are marked using the track change view of MS Word.

In brief, we used 'race' instead of 'ethnicity' as appropriate, lowered the tone of conclusion due to the heterogeneity among the studies, revised the graphical abstract spelling mistake, provided possible explanations for the lower role of BMI in GC risk among Asians, and improved Introduction and Discussion by adding mentioned points about cardia and not cardia GC and dietary factors.

Comment 1: Regarding the association between BMI and race, Asian race seems to be less influenced by obesity and overweight in the manuscript. The Asian populations are known to have the highest rate of GC in the world and there are also screening programs for GC. Please, comment on these two points: how the highest prevalence of GC overall can determine the lower role of BMI and how the available screening (Endoscopy, serological tests...) could influence the prevalence of GC and therefore create a big bias in the evaluation of the role of BMI between Asian and no Asian populations. There are a lot of studies on these topics. It would be interesting to adjust the analysis, if possible.

R2-C1-Response: Thank you for your review and insightful comment. One possible explanation for the lower role of BMI in GC risk among Asians is that other risk factors, such as Helicobacter pylori infection, genetics, and high-sodium diet, may play a more prominent role in the development of GC in this population. Also, the lower association between BMI and GC in Asians reported in the current study may be due to the fact that Asians tend to have a higher proportion of body fat and a lower proportion of muscle mass compared to other populations at the same BMI level. In 2004, the World Health Organization reviewed the evidence showing that Asians have a higher risk of developing weight-related diseases at lower BMIs. However, due to a lack of agreement among researchers, it did not establish distinct BMI cutoffs for this population. In recent years, further studies have led some groups to adopt different BMI and abdominal obesity thresholds for Asians and even for different Asian groups. Research has shown that Asian individuals tend to accumulate weight around the central region of their bodies and at a faster rate than other ethnic groups. Studies have shown that even when age, BMI, and total fat mass are taken into account, Chinese and South Asian individuals have significantly higher levels of visceral adipose tissue compared to White individuals. Therefore, to assess the relationship between GC and body composition, especially in the Asian population, considering other indicators such as waist circumference and waist-to-hip ratio may be necessary. (We added this paragraph to Discussion, Lines 314-334.)

Additionally, a recent systematic review and meta-analysis by Zhang et al. showed that endoscopic screening for GC can influence the prevalence of GC by detecting cases at earlier stages, resulting in lower mortality rates. However, the incidence of gastric cancer in Asian countries was not found to be significantly affected by screening. On the other hand, none of the studies included in our analysis reported screening programs as a confounding factor or adjusted for it. Therefore, we are unable to adjust the results based on the effect of screening programs. One way to determine the impact of screening programs would be to conduct a subgroup analysis based on the region of study that has implemented national screening programs. However, except for one study (Jang et al.), which reported cumulative results from different countries, including Singapore, Korea, Japan, and China, the other Asian countries included in our manuscript are the same as countries with national screening programs (Japan, Korea, and China). We conducted a sensitivity analysis by excluding the Jang et al. study, which did not meaningfully alter the results.

Comment 2: There are several studies on the role of meat, salt and diet in general and GC. Please, add them in the introduction and in the discussion.

R2-C2-Response: Thanks for pointing this out. We added the mentioned points to Introduction (lines 95-99) and Discussion (lines: 373-375)

Comment 3: In the discussion: from the line 358 to 370 the authors talk about obesity in general. This part should belong to the introduction.

R2-C3-Response: Thanks for your time on reading and evaluating the manuscript. We eliminated extra information and added essential findings of this paragraph into Introduction, second paragraph, lines 77-81.

Comment 4: Still I think that the discussion about cardia and not cardia GC is little and should be improved. Indeed, the discussion should be implemented with studies on the association between obesity and MRGE and not-cardia GC, if possible.

R2-C4-Response: Thank you for bringing up this valuable point. We added the mentioned point to Discussion section, lines 353-367.

The authors want to again appreciate the time and effort that reviewers and editors put into evaluating this study. The comments improved the manuscript and analysis, and any further comments would be welcomed.

Bests,

Mohammad Javad Nasiri (mj.nasiri@hotmail.com)

Round 3

Reviewer 2 Report

Dear authors, the text appears improved and modified in a better understandable and more complete way. However, some main topics are lacking, as I underlined in the previous report to your manuscript.

Indeed, the difference of prevalence of GC between Asian and non-asian population have to be stressed. This is a crucial point, as it is one of the reason why the BMI is less influent in Asia than in western countries. Indeed, the higher prevalence forced the Asian countries to have screening programs for the detection of early lesions. This changes the founding of advanced lesions and, therefore, changes the relationship between GC and any other factors, such as BMI. It would be interesting to include in the multivariate analysis the presence of a screening program for GC in the country of the study, to see if the relationship between BMI and GC is stronger in the countries without screening program or not. Indeed, it could be a big bias, because many patients benefit of the removal of early lesions and thus, they are not included in the GC group.

Author Response

Dear reviewer,

The authors want to appreciate the time and effort that you provided to give concise feedback and comments for our manuscript entitled “Gastric Cancer Risk in Association with Underweight, Overweight, and Obesity: A Systematic Review and Meta-Analysis”. We improved the manuscript and corrected flaws using your comments. All changes are marked using the track change view of MS Word.

Comment 1: Dear authors, the text appears improved and modified in a better understandable and more complete way. However, some main topics are lacking, as I underlined in the previous report to your manuscript.

Indeed, the difference of prevalence of GC between Asian and non-asian population have to be stressed. This is a crucial point, as it is one of the reason why the BMI is less influent in Asia than in western countries. Indeed, the higher prevalence forced the Asian countries to have screening programs for the detection of early lesions. This changes the founding of advanced lesions and, therefore, changes the relationship between GC and any other factors, such as BMI. It would be interesting to include in the multivariate analysis the presence of a screening program for GC in the country of the study, to see if the relationship between BMI and GC is stronger in the countries without screening program or not. Indeed, it could be a big bias, because many patients benefit of the removal of early lesions and thus, they are not included in the GC group.

R2-C1-Response: Thank you for your valuable feedback. We appreciate your suggestion to adjust our results for screening programs as this can have a significant impact on the relationship between BMI and GC, especially in Asian countries with high prevalence of GC. However, as a major limitation of meta-analysis, we are unable to adjust our results for factors that were not reported or measured in the included studies. Regrettably, none of the studies that met our inclusion criteria provided adjusted data based on screening programs. Therefore, we discussed the impact of screening programs on the relationship between BMI and GC in Asian and non-Asian countries and acknowledged this potential bias as a limitation of our study. (Lines: 335-343, and 409-415)

The authors want to again appreciate the time and effort that reviewers and editors put into evaluating this study. The comments improved the manuscript and analysis, and any further comments would be welcomed.

Bests,

Mohammad Javad Nasiri (mj.nasiri@hotmail.com)

Round 4

Reviewer 2 Report

Thank you for the improvement. 

Please, add a short but complete chapter with the limitations of the study at the end of the paper

Author Response

Dear reviewer,

The authors want to appreciate the time and effort that you provided to give concise feedback and comments for our manuscript entitled “Gastric Cancer Risk in Association with Underweight, Overweight, and Obesity: A Systematic Review and Meta-Analysis”. We improved the manuscript and corrected flaws using the editor’s and reviewers’ comments. Here is a point-by-point response to the reviewers’ comments and concerns (the address of change is presented at the end of the response, in the parenthesis). All changes are marked using the track change view of MS Word.

In brief, we acknowledged a limitation of our meta-analysis, which is the potential impact of confounding factors on the relationship between BMI and gastric cancer.

Comment 1: Please, add a short but complete chapter with the limitations of the study at the end of the paper.

R2-C1-Response: Thank you for your feedback. We appreciate your suggestion and added this statement as a limitation of the meta-analysis: “A meta-analysis of individual patient data may be warranted in the future since a meta-analysis itself cannot adjust for the effect of confounding factors.” (Lines: 417-418). So the second part of the limitation, completely discusses the limitation of meta-analysis in addressing the role of confounding factors as follows:

“(2) Several potential confounding factors, including physical activity, dietary intake, H. pylori infection, and screening programs, could influence the results of the included observational studies. The presence of screening programs in Asian countries with high GC prevalence may alter the association between GC and other factors such as BMI. This could introduce bias since many patients benefit from early lesion removal and thus may not be included in the GC group. A meta-analysis of individual patient data may be warranted in the future since a meta-analysis itself cannot adjust for the effect of confounding factors.” (Lines: 409-419)

The authors want to again appreciate the time and effort that reviewers and editors put into evaluating this study. The comments improved the manuscript and analysis, and any further comments would be welcomed.

Bests,

Mohammad Javad Nasiri (mj.nasiri@hotmail.com)
